# Mental health of pregnant women during the SARS-CoV-2 pandemic in France: Evolution of self-perceived psychological state during the first lockdown, and anxiety frequency two months after the lockdown ended

**Alexandra Doncarli**[1☯*], **Lucia Araujo-Chaveron**[1☯], **Catherine Crenn-Hebert**[2,3], **Marie-Noëlle Vacheron**[4], **Christophe Léon**[1], **Imane Khireddine**[1], **Francis Chin**[1], **Alexandra Benachi**[5,6], **Sarah Tebeka**[1,7,8], **Nolwenn Regnault**[1]

1 Santé Publique France, Saint-Maurice, France, 2 Department of Gynecology and Obstetrics, Louis Mourier University Hospital, AP-HP, Colombes, France, 3 Regional Health Agency of Ile de France (ARS-IDF), Saint-Denis, France, 4 Consultation d'Information, Conseils et Orientation Pour les Femmes Enceintes ou Avec Désir D'Enfant Atteintes de Trouble Psychique (CICO), GHU Paris Psychiatrie et Neurosciences, Hôpital Saint-Anne, Paris, France, 5 Department of Obstetrics and Gynecology, Antoine Beclere Hospital, AP-HP, Clamart, France, 6 Université Paris Saclay, Clamart, France, 7 Université Paris Cité, INSERM U1266, F-75014, Paris, France, 8 Department of Psychiatry, AP-HP, Louis Mourier Hospital, F-92700, Colombes, France

☯ These authors contributed equally to this work.
\* alexandra.doncarli@santepubliquefrance.fr

**Data Availability Statement:** The data set contains individual data potentially identifying or sensitive

## Abstract

Previous pandemics and related lockdowns have had a deleterious impact on pregnant women's mental health. We studied the impact of the SARS-CoV-2/Covid-19 pandemic and France's first lockdown on pregnant women's mental health. A cross-sectional study was conducted in July 2020 using a web-questionnaire completed by 500 adult women who were pregnant during the first lockdown in France (March-May 2020). Questions focused on their self-perceived psychological state and affects they felt before and during the lockdown and anxiety symptomatology (HAD) two months after it ended. A robust variance Poisson regression model was used to estimate adjusted prevalence ratios (aPR) for anxiety and self-perceived psychological state evolution. One in five respondents (21.1%) reported psychological deterioration during lockdown. Associated determinants were: i) little or no social support (self-perceived) (aRP = 1.77, 95%CI[1.18–2.66]), ii) increased workload (1.65, [1.02–2.66]), and iii) poor/moderate knowledge about SARS-CoV-2 transmission (1.60, [1.09–2.35]). Seven percent of women reporting psychological deterioration had access to professional psychological support during lockdown, while 19% did not despite wanting it. Women reported heightened powerlessness (60.3%), frustration (64%) and fear (59.2%) during lockdown. One in seven respondents (14.2%, 95%CI[10.9–18.2]) had anxiety symptoms. Determinants associated: i) at least one pregnancy-related pathology (aPR = 1.82, 95%CI[1.15–2.88]), ii) overweightness or obesity (1.61, [1.07–2.43]), iii) one child under the age of six years in the household during the lockdown (3.26, [1.24–8.53]), iv) little or no social support (self-perceived) during the lockdown (1.66, [1.07–2.58]), v) friend or

patient information (childbirth's data, age, parity, region of residence, number and age of other children, violence, Coronavirus infection status, chronic diseases, etc.). It cannot therefore be shared publicly. However, researchers who meet the criteria for access to confidential data can request access to these data by writing to DATA-MAD@santepubliquefrance.fr.

**Funding:** The author(s) received no specific funding for this work.

**Competing interests:** The authors have declared that no competing interests exist.

relatives diagnosed with Covid-19 or with symptoms of the disease (1.66; [1.06–2.60]), vi) no access to medication for psychological distress (2.86, [1.74–4.71]), and vii) unsuccessfully seeking exchanges with healthcare professionals about their pregnancy during the pandemic (1.66, [1.08–2.55]). Our results can guide prevention and support policies for pregnant women during pandemics, current or future, with or without lockdowns. Preventing perinatal mental health problems is essential to ensure a supportive environment for the child's development.

## Introduction

Data from previous coronavirus outbreaks in 2002 and 2013 showed that pregnancy was a risk factor for severe forms of associated respiratory diseases. More specifically, SARS-CoV-1 and MERS-CoV were associated with significant acute respiratory distress syndrome [1,2]. This reality, together with recommendations of learned societies [3], prompted several countries, including France, to declare in March/April 2020 that pregnant women should be considered a population at greater risk of severe forms of Covid-19, the disease caused by SARS-CoV-2 [4–7]. In the absence of vaccines and effective pharmaceutical treatments at that time, most governments decided to reduce the spread of the virus by implementing strict lockdowns of their entire population for several months. In France, the first such lockdown took place between 17 March and 11 May 2020.

Recent studies showed the negative psychological effects of lockdowns implemented during previous epidemics, including anxiety disorders, depressive disorders, psychological distress and sleep disorders [8,9]. Some of these negative psychological effects seem to persist after the lockdown period [9]. Furthermore, the lockdown measures implemented to prevent the virus' diffusion might have led to social deprivation and a lack of sufficient social support (i.e., from family, friends, etc.), two known risk factors for mental health fragility in women during the perinatal period [10,11]. Thus, there was a potential psychological impact of lockdown measures in pregnant women.

In addition, pregnant women—whether they were infected or not—knew that they were at greater risk of developing severe forms of Covid-19, and consequently may have felt worried about their own health and especially that of their unborn or newborn child. Thus, although little studied at the beginning of the pandemic (this is also true for the previous SARS-CoV-1, MERS-CoV, and H1N1 outbreaks [12,13]), there was a potential psychological impact linked to the perception of the risk of developing a severe form of Covid-19 in pregnant women. And this could be a cumulative impact with lockdown measures on the mental health of pregnant women. Several recent studies showed indeed the negative psychological impact of the ongoing SARS-CoV-2 pandemic—whose scale and duration are unprecedented—on anxiety, depression, and hostility in pregnant women. A Quebec study on pregnant women before (n = 496) and during (n = 1258) the pandemic reported a higher level of depressive, anxiety symptoms, and negative affect, and less positive affectivity [14]. Furthermore, a Chinese study reported higher scores of depression, anxiety and hostility, as well as sleep disturbances in pregnant women [15].

Psychological disorders related to the current pandemic might also negatively impact pregnant women's general physical health and their child's physical and mental health, especially in terms of the risk of premature birth and low child birth weight. Most of all, they negatively impact the development of the mother-newborn bond [16–19].

Given this context, we designed and implemented Covimater, a population-based study whose objectives were (i) to explore the evolution in self-perceived psychological state between before and during the first French lockdown (March-May 2020) in a sample of pregnant women, (ii) to assess anxiety frequency and factors associated with anxiety symptoms in these women two months after the end of lockdown.

## Materials and methods

### Study design, setting and sample size of Covimater

At our request, a service provider (BVA group) interviewed its unpaid pre-pandemic internet panel of 15,000 future parents or parents of children under 3 years to create a pseudonymised non-probabilistic sample of 500 pregnant adult women who met the inclusion criteria (described below) and volunteered to participate in our survey. Covimater is a cross-sectional study using quotas sampling, whereby the study sample is assigned a structure similar to that of the target population (i.e., all pregnant women) in order to tend towards representativeness. The population of parents of children under 1 year old—as per the National Institute of Statistics and Economic Studies 2016 census—was used to set the quotas [20]. By its broad representation, the latter was judged a good proxy for our target population of pregnant women in France. The quotas for mothers of children under 1 year old were applied to calculate weightings using Newton's algorithm [21] and obtain weighted individual data for the statistical analysis presented herein (see below). Specifically, these quotas comprised age group, socio-professional category (SPC), region of residence, size of urban area, and parity.

Eligible women (see below) were invited by BVA to answer an online questionnaire between 6 and 20 July 2020 i.e. two months after the end of the first lockdown in France (March-May 2020). The two-month interval was chosen i) to avoid the memory bias associated with a longer interval, and ii) because the major recommended prevention measures had not changed in the two months after lockdown. No significant difference in available data for age group, region of residence, or parity was observed between the women participating in Covimater and women in the whole French population who gave birth in a hospital maternity ward (i.e., 99% of pregnant women in France [22]).

### Participants

Our sample comprised 500 women who were: i) pregnant during the first lockdown in France (from 17 March to 11 May 2020), ii) aged 18 and over, and iii) residents in metropolitan France. We excluded two groups of women pregnant during lockdown but with limited exposure to it: those who delivered in the two first weeks of lockdown and those whose first week of gestation began during the last two weeks of lockdown (deducted from the expected date of delivery reported by the women).

### Issues of interest: Mental health measures

Three aspects were studied:

- Change in self-declared general psychological state was assessed with the following two questions: "Just before the lockdown, on a psychological level, how did you feel?" (good/quite good/quite poor/poor), and "During the lockdown, on a psychological level, how did you feel?" (good/quite good/quite poor/poor). In our analysis, the first study outcome was psychological deterioration defined as a switch from a 'good' or 'quite good' psychological state just before the lockdown to 'quite poor' or 'poor' state during the lockdown.

- Positive and negative study affects felt more strongly than usual during the lockdown: relief, serenity, security, loneliness, frustration, powerlessness, anger, fear and despair. The related question was: "During lockdown, did you feel the following emotions more strongly than usual? (Yes, a lot/Yes, somewhat/No, not really/No, not at all). In the analysis, for each affect, it was estimated that women who answered "yes, a lot" or "yes, somewhat" felt that affect more strongly than usual during the lockdown.

- Anxiety symptoms two months after the first lockdown ended. Women were screened using answers to the seven questions on anxiety in the 14-item Hospital Anxiety and Depression scale (HAD) [23]. A score from 0 to 3 is assigned to each HAD question. Women with an overall score of >10 for all seven anxiety questions were considered to have anxiety symptoms [24].

## Comparisons

Explanatory variables were divided into five main themes:

Demographic and socio-economic: age, socio-professional category (SPC) reduced into SPC+ (self-employed women, managers, intermediate professions), SPC- (employees, blue-collar workers) and inactive women (students and other professionally inactives), educational level (equal to or higher than secondary school diploma, lower than secondary school diploma), perceived financial situation (comfortable, just getting by, difficult to make ends meet).

Pandemic and lockdown-related: child(ren) under six years of age (*i.e.*, younger than required school age in France) in the household during the lockdown, SARS-CoV-2 strain-on healthcare system in region of residence (coded as green, orange or red, reflecting increased epidemic pressure) [25], professional workload (did not work, lighter than/same as usual, heavier than usual), self-perceived social support (from family, friends, etc.; Very good/Good, Little or None), experience of serious disputes/climat of violence (Very-often/Often, Some-times/Rarely, Never), level of knowledge about the virus' modes of transmission (score based on seven questions, see details in Table 1), presence of COVID-19-type symptoms, family member or friends with COVID-19 diagnosis or symptoms suggestive of the disease.

Self-perception of the pandemic during the lockdown: A scale-based score was recorded for participants' perceived vulnerability to SARS-CoV-2 infection (from 0 (not at all vulnerable) to 10 (vulnerable)). A dichotomous variable was then created with 6/10 as the thresholds corresponding to the average vulnerability observed (6.2 +/- 0.1).

Pregnancy and health: parity, gestational age at the end of the first lockdown, childbirth (during or after first lockdown), at least one pre-existing chronic disease or pregnancy-related pathology (see details of pathologies in Table 1, notes f and g), overweight/obesity status before pregnancy (based on body mass index$\geq$25kg/m$^2$; see Table 1, note h).

Pregnancy monitoring during first lockdown: had a consultation/examination cancelled/postponed on a health professional's initiative, unsuccessful attempts to have an exchange with healthcare professionals during lockdown about their pregnancy and the pandemic (women who reported not needing such an exchange and those who did talk to a professional were considered to have had successful attempts), taking medication for mood disorders or sleeping problems during the lockdown.

More details are described in others publications about Covimater ([26–28]).

## Ethics and endpoint

Covimater received approval from the Saint Maurice Hospital Ethics Committee on 01/07/2020 (approval number n°2020–1). Internet panel volunteers of adult women (>18 years)

**Table 1. Description of pregnant women during the first COVID-19-related lockdown (March-May 2020) who participated in the Covimater survey (n = 500), France (July 2020).**

| | N (%) or mean (sd)* | | [95%CI**] |
|---|---|---|---|
| **Demographic and socio-economic characteristics** | | | |
| Age (in years) | 31.4 | (5.1) | [30.8–31.9] |
| Socio-professional category (SPC)[a] | | | |
| SPC + | 192 | (38.4) | [33.9–43.2] |
| SPC - | 180 | (36.1) | [31.8–40.6] |
| Inactive | 128 | (25.5) | [20.5–31.2] |
| Educational level | | | |
| Equal to or higher than secondary school diploma | 391 | (78.1) | [73.6–82.1] |
| Lower than secondary school diploma | 109 | (21.9) | [17.9–26.4] |
| Perceived financial situation | | | |
| Comfortable | 246 | (49.2) | [44.2–54.2] |
| Just getting by | 159 | (31.7) | [27.2–36.6] |
| Difficult to make ends meet | 95 | (19.1) | [15.2–23.7] |
| **Pandemic and lockdown related variables** | | | |
| Child(ren) under six years of age in the household during the lockdown | 234 | (46.8) | [41.8–51.8] |
| SARS-CoV-2 strain on healthcare system (colour-coded) for the region of residence[b] | | | |
| Green zone | 127 | (25.4) | [21.1–30.2] |
| Orange zone | 150 | (30.0) | [25.7–34.7] |
| Red zone | 223 | (44.6) | [39.7–49.6] |
| Professional workload | | | |
| Did not work | 351 | (70.1) | [65.7–74.2] |
| Lighter or same as usual | 85 | (17.1) | [14.0–20.7] |
| Heavier than usual | 64 | (12.8) | [10.1–16.0] |
| Self-perceived social support | | | |
| Very good / Good | 411 | (82.1) | [78.2–85.5] |
| Little or none | 89 | (17.9) | [14.5–21.8] |
| Serious disputes/climat of violence | | | |
| Very-often/ Often | 11 | (2.3) | [1.10–4.60] |
| Sometimes / Rarely | 129 | (25.8) | [21.7–30.4] |
| Never | 360 | (71.9) | [67.2–76.2] |
| Level of knowledge about SARS-CoV-2 transmission[c] | | | |
| Good knowledge | 170 | (34.0) | [29.3–38.9] |
| Poor/Moderate knowledge | 330 | (66.0) | [61.1–70.7] |
| Experiencing COVID-19 type symptoms | 92 | (18.4) | [14.9–22.6] |
| Family member or friend with COVID-19 diagnosis or symptoms suggestive of the disease | 171 | (34.2) | [29.7–39.0] |
| **Self-perception of the pandemic during first lockdown** | | | |
| Perceived vulnerability to severe forms of COVID-19 disease (max. 10; n = 459) >6/10[d] | 250 | (54.6) | [49.4–59.6] |
| **Pregnancy and health** | | | |
| Primiparous | 203 | (40.6) | [35.8–45.6] |
| Gestational age (weeks)[e] | | | |
| <10 | 34 | (6.8) | [4.70–9.80] |
| 10–20 | 177 | (35.4) | [30.8–40.3] |
| 20–30 | 180 | (36.1) | [31.4–41.0] |
| 30–40 | 77 | (15.4) | [12.1–19.4] |
| > 40 | 32 | (6.3) | [4.30–9.20] |

(*Continued*)

**Table 1.** (Continued)

| | N (%) or mean (sd)* | | [95%CI**] |
|---|---|---|---|
| Childbirth | | | |
| During lockdown | 34 | (6.8) | [4.70–9.80] |
| After lockdown | 466 | (93.2) | [90.2–95.2] |
| Pre-existing chronic disease(s)[f] | 152 | (30.3) | [25.8–35.1] |
| Pregnancy-related pathology(ies)[g] | 119 | (23.7) | [19.9–28.0] |
| Overweight/obesity status before pregnancy[h] | 212 | (42.4) | [37.5–47.4] |
| **Pregnancy monitoring during first lockdown** | | | |
| Cancelled/postponed pregnancy consultations or examinations at the initiative of a health professional | 182 | (36.3) | [31.6–41.3] |
| Having an unmet need to communicate with health professionals about course of pregnancy/childbirth during pandemic | | | |
| No | 295 | (59.0) | [53.9–63.8] |
| Yes | 205 | (41.0) | [36.1–46.1] |
| Took medication for mood disorders/sleeping disorders | | | |
| No, because I did not need it | 456 | (91.2) | [87.9–93.7] |
| Yes | 20 | (3.9) | [2.3–6.5] |
| No, but I would have liked to | 24 | (4,9) | [3.1–7.5] |

* Weighted and rounded values using Newton's algorithm [21] for discrete or qualitative variables. For continuous variables, mean (standard deviation) were presented.

** 95% Confidence Interval.

[a] Women on maternity leave and unemployed women were classified according to their current SPC category or their most recent category prior to ending work, respectively.

[b] Estimated by the Ministry of Health on 1 May 2020 on the basis of two variables: i) Virus circulation level (i.e., percentage of emergency room admissions for suspected COVID-19) and ii) Strain on hospital intensive care unit capacity (i.e., occupancy rate of intensive care beds by patients with COVID-19), coded as green, orange or red, reflecting increased epidemic pressure on the healthcare system [25].

[c] Score based on seven questions (Good knowledge if all answers were correct; Poor/moderate knowledge else).

[d] Scores for participants' perceived vulnerability to SARS-CoV-2 infection during the first lockdown (from 0 (not at all vulnerable) to 10 (very vulnerable)). A dichotomous 'low/high' variables were then created for 'vulnerability', with 6/10 as the thresholds (see details in methods). No documented data for 41 pregnant women in terms of level of perceived vulnerability to severe forms of COVID -19.

[e] At the end of the first lockdown (11 May 2020).

[f] Diabetes, Overweight/obesity status before pregnancy, High blood pressure, Asthma, Cardiac condition, Autoimmune disease, Mental illness, Inherited bleeding disorders.

[g] Gestational diabetes, Pre-eclampsia, Preterm labour, Gestational hypertension.

[h] Body mass index$\geq$25kg/m$^2$.

included in the Covimater study were informed by mail of the study's purpose then given the choice by mail to participate in the survey. Only pseudonymised databases were transmitted to Santé publique France. The data are stored on Santé publique France's servers, respecting the agency's data security and confidentiality standards.

## Statistical analysis

A robust variance Poisson regression model was used to estimate unadjusted and adjusted prevalence ratios (aPR) [29] for two of the three study outcomes: declared psychological

deterioration and anxiety symptoms. Factors associated with each of these two outcomes which either had a p-value<0.20 in bivariate analysis or were judged to be clinically relevant based on the literature (gestational age at completion of study questionnaire, gestational age at end of lockdown period, parity), were introduced into the multivariate models. When several variables were possibly collinear, the model with the best likelihood score (lowest Bayesian information criterion) was selected. Fractional polynomials showed a linear relationship between continuous variables included in the models and the studied prevalence of each of the two outcomes. A manual descending stepwise procedure was then applied to identify factors independently associated (p-value<0.05) with each outcome. Hosmer-Lemeshow tests were performed to verify the goodness of fit of each final model. Estimates of aPR, their 95% confidence intervals (95%CI) and associated p-values are presented. As indicated by Zou, PRs can be interpreted in the same way as relative risk (RR) [30].

Only a descriptive analysis was presented for women who felt affects more strongly than usual during the lockdown (percentage and related 95%CI). The average affects (+/- standard deviation (sd)) experienced during the lockdown was also calculated for the six negative affects (frustration, anger, powerlessness, despair, loneliness and fear).

All statistical analyses were performed using Stata software ®version 14.2 (Stata Corp., College Station, TX).

## Results

### Characteristics of women included in Covimater (Table 1)

Mean age was 31.4 years (sd = 5.1). Four-fifths (78.1%) had a level of education equivalent to or higher than secondary school diploma, 36.1% were classified SPC-, 25.5% were inactive, 31.7% declared they just got by financially while 19.1% reported that they could not make ends meet. Regarding their area of residence, 44.6% lived in a red-coloured zone (i.e., highest pressure on the healthcare system). Less than half had a child under six years of age in their home at the time of the lockdown. Just over a third had family members or friends who were diagnosed with Covid-19 or had symptoms suggestive of the disease. Finally, 17.9% perceived little or no social support during the lockdown, and nearly 28.1% had experienced serious dispute or a climate of violence.

### Psychological state between before and during the first lockdown (Table 2 and Fig 1)

More than half (52.8%) of the study sample reported a poorer psychological state between before the lockdown and during it.

Furthermore, 21.1% had a marked deteriorated psychological state (i.e., they self-reported that their pre-lockdown state was good/quite good but their state during lockdown was quite poor/poor). This deterioration was higher among those who felt they received little or no social support (aPR = 1.77, 95%CI[1.18–2.66]), those with a heavier than usual workload (1.65 [1.02–2.66]), and those with poor/moderate knowledge of the symptoms and modes of transmission of SARS-CoV-2 (1.60, [1.09–2.35]) (Table 2). Furthermore, of the women who reported a psychological deterioration during the lockdown, 7% consulted a psychiatrist or psychologist for psychological support during the lockdown, 19% wanted such support but did not get it, while 74% reported they did not need it (p<0.0001).

A majority of respondents reported experiencing more intense feelings of powerlessness (60.3%), frustration (64.0%), and fear (59.2%) than usual during the lockdown (Fig 1). They

**Table 2. Factors associated with psychological deterioration felt during lockdown, Covimater survey (n = 500), France (July 2020).**

| | Obs. (n %)* | | Psychological deterioration felt (n = 105)[a] | | |
| --- | --- | --- | --- | --- | --- |
| | | | Yes (n %)* | | Adjusted PR [95% CI] ** | p-value** |
| **Demographic and socio-economic characteristics** | | | | | | |
| Age (in years) | 31.4 | (5.1) | 31.6 | (5.2) | 1.00 [0.96–1.04] | 0.74 |
| Gestational age (in weeks)[b] | 23.5 | (9.0) | 24.2 | (9.7) | 1.01 [0.99–1.03] | 0.32 |
| Socio-professional category[c] | | | | | | |
| SPC+ | 192 | (38.4) | 37 | (19.3) | 1 | |
| SPC- | 180 | (36.1) | 39 | (21.7) | 1.11 [0.74–1.67] | 0.60 |
| Inactive | 128 | (25.5) | 29 | (22.6) | 1.23 [0.66–2.28] | 0.51 |
| Parity | | | | | | |
| Primiparous | 203 | (40.6) | 42 | (20.7) | 1 | |
| Multiparous | 297 | (59.4) | 63 | (21.2) | 1.09 [0.70–1.71] | 0.70 |
| **Pandemic and lockdown related variables** | | | | | | |
| Self-perceived social support | | | | | | |
| Very good / Good | 411 | (82.1) | 75 | (18.2) | 1 | |
| Little or none | 89 | (17.9) | 30 | (33.7) | **1.77 [1.18–2.66]** | **0.006** |
| Workload during lockdown | | | | | | |
| Did not work | 351 | (70.1) | 72 | (20.5) | 1 | |
| Lighter or same as usual | 85 | (17.1) | 14 | (16.5) | 0.98 [0.58–1.65] | 0.94 |
| Heavier than usual | 64 | (12.8) | 19 | (29.7) | **1.65 [1.02–2.66]** | **0.04** |
| Knowledge about modes of transmission of SARS-CoV-2[d] | | | | | | |
| Good knowledge | 170 | (34.0) | 57 | (17.3) | 1 | |
| Poor/moderate knowledge | 330 | (66.0) | 48 | (28.2) | **1.60 [1.09–2.35]** | **0.02** |

* Weighted and rounded values using Newton's algorithm for discrete or qualitative variables [21]. For continuous variables (age, pregnancy term), mean (standard deviation) were presented.

** Adjusted Prevalence Ratio (aPR), Confidence Interval 95% (95%CI) and p-value obtained with robust variance Poisson regression model.

[a] Psychological deterioration defined as a switch from a 'good' or 'quite good' psychological state just before the lockdown to 'quite poor' or 'poor' state during the lockdown. (see definition of the variable of interest in Methods section).

[b] At the end of the first lockdown (11/05/2020).

[c] Women on maternity leave and unemployed women were classified according to their current SPC category or their most recent category prior to ending work respectively.

[d] Score based on seven questions (Good knowledge if all answers were correct; Poor/moderate knowledge else).

also reported an average of 2.97 (sd = 2.94–2.99) negative affects out of the six proposed (frustration, anger, powerlessness, despair, loneliness and fear) during the lockdown.

For each affect, women were asked: "During lockdown, did you feel this emotion more strongly than usual? (Yes, a lot/Yes, somewhat/No, not really/No, not at all). Women who answered "yes, a lot" or "yes, somewhat" were considered to have felt this affect more strongly than usual during the lockdown. This percentage and the 95% confidence interval (95%CI) were calculated and presented here.

## Anxiety symptoms in study sample two months after the first French lockdown ended (Table 3)

Among women who took part in the survey, 14.2% (IC95%[10.9–18.2]) had anxiety symptoms. After adjusting for age, gestational age, SPC, parity and depression score, factors associated with having anxiety symptoms were as follows: at least one pregnancy-related pathology

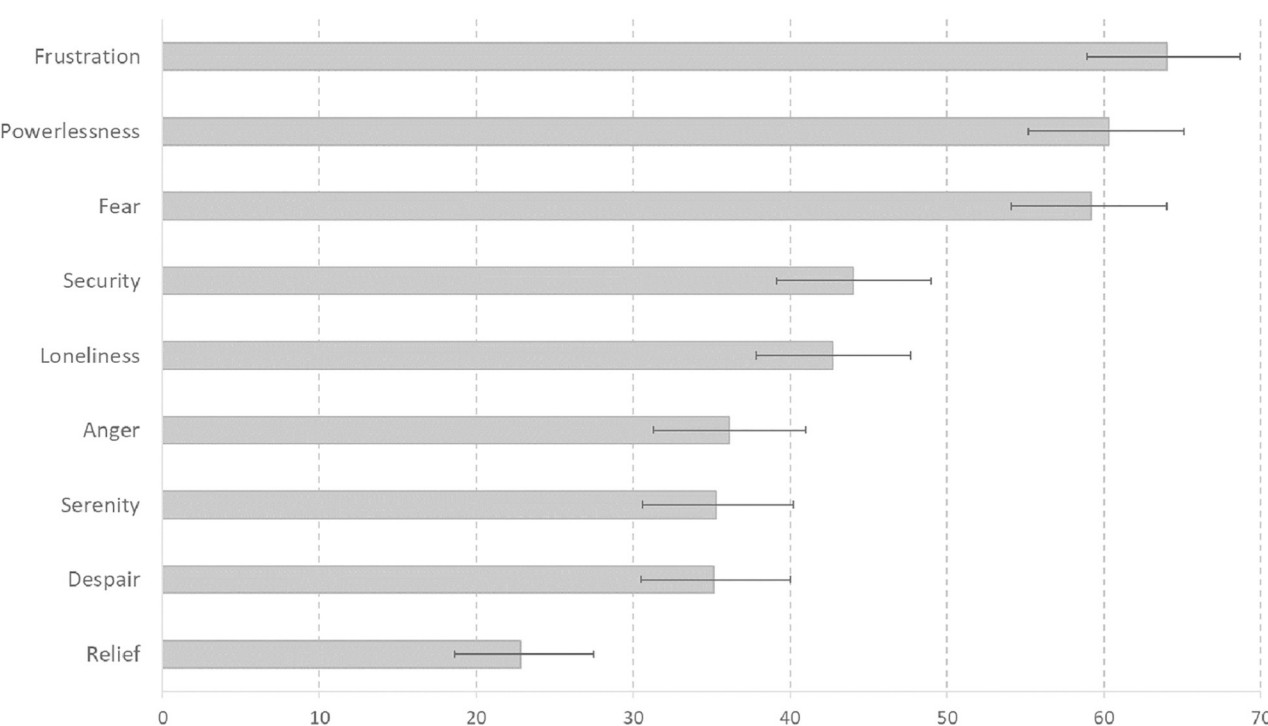

**Fig 1. Affects felt more strongly than usual during lockdown (March-May 2020) in pregnant women who participated in the Covimater survey (n = 500), France (July 2020).**

(aPR = 1.82, 95%CI[1.15–2.88]), overweightness or obesity (1.61, [1.07–2.43]), at least one child under the age of six years in the household during the lockdown (3.26, [1.24–8.53]), self-perceiving little or no social support during the lockdown (1.66, [1.07–2.58]), and having loved ones diagnosed with Covid-19 or with symptoms suggestive of the disease (1.66, [1.06–2.60]). In addition, anxiety symptoms were more frequent in women who unsuccessfully tried to have an exchange with a healthcare professional about impact of the pandemic on their pregnancy, than in those who had such an exchange (1.66, [1.08–2.55]). Similarly, anxiety symptoms were more frequent in women who unsuccessfully tried to obtain medication for mood disorders or sleeping disorders (2.86, [1.74–4.71]) than in those who responded that they did not take medication because they did not need it. However, the frequency of anxiety symptoms was not significantly different between women who had access to medication and those who did not (p = 0.248).

## Discussion

This study assessed the some aspects of mental health of pregnant women living in France during the ongoing SARS-CoV-2 pandemic. Of the 500 women in the study sample, 52.8% declared a poorer mental health during lockdown than before it. More specifically, 21.1% were defined as having a deteriorated psychological state (i.e., they self-reported that their pre-lockdown state was good/quite good but their state during lockdown was quite poor/poor). Factors associated with this drastic deterioration in mental health during the lockdown were perceiving little or no social support (i.e., family, friends, etc.) and a heavier-than-usual workload. In contrast, having a good level of knowledge about the modes of transmission of the SARS-CoV-

**Table 3. Factors associated with anxiety symptoms frequency (defined as an HAD score>10) two months after the first SARS-CoV-2 pandemic lockdown ended in pregnant women who participated in the Covimater survey (n = 500), France (July 2020).**

| | Obs. (n %)* | | Anxiety symptoms (HAD score >10); n = 71[a] | | |
| --- | --- | --- | --- | --- | --- |
| | | | Yes (n %)* | | Adjusted PR [95% CI]** | p-value** |
| **Demographic and socio-economic characteristics** | | | | | |
| Age (in years) | 31.4 | (5.1) | 30.7 | (5.4) | **0.96 [0.92–0.99]** | **0.03** |
| Gestational age (in weeks)[b] | 23.5 | (9.0) | 31.8 | (7.1) | 1.01 [0.98–1.04] | 0.63 |
| Socio-professional category[c] | | | | | | |
| SPC+ | 192 | (38.4) | 25 | (13.0) | 1 | |
| SPC- | 180 | (36.1) | 23 | (12.8) | 0.86 [0.53–1.40] | 0.56 |
| Inactive | 128 | (25.5) | 23 | (17.9) | 0.85 [0.47–1.53] | 0.59 |
| Parity | | | | | | |
| Primiparous | 203 | (40.6) | 29 | (14.3) | 1 | |
| Multiparous | 297 | (59.4) | 42 | (14.1) | 0.54 [0.21–1.36] | 0.19 |
| **Pregnancy and health** | | | | | |
| Pregnancy-related pathology[d] | | | | | | |
| No | 411 | (82.1) | 42 | (11.0) | 1 | |
| Yes | 89 | (17.9) | 29 | (24.3) | **1.82 [1.15–2.88]** | **0.01** |
| Overweight/Obesity[e] | | | | | | |
| No | 351 | (70.1) | 32 | (11.1) | 1 | |
| Yes | 85 | (17.1) | 39 | (18.4) | **1.61 [1.07–2.43]** | **0.02** |
| Depression symptoms (HAD score >10) | 64 | (12.8) | 9.1 | (3.8) | **1.19 [1.13–1.25]** | **<0.0001** |
| **Pandemic and lockdown related variables** | | | | | |
| Child(ren) under the age of six years of age in the household | | | | | | |
| No | 170 | (34.0) | 27 | (10,1) | 1 | |
| Yes | 330 | (66.0) | 44 | (18.8) | **3.26 [1.24–8.53]** | **0.02** |
| Self-perceived social support | | | | | | |
| Very good / Good | 411 | (82.1) | 45 | (10.9) | 1 | |
| Little or none | 89 | (17.9) | 26 | (29.2) | **1.66 [1.07–2.58]** | **0.02** |
| Family member or friend with COVID-19 diagnosis or symptoms suggestive of the disease | | | | | | |
| No | 329 | (65.8) | 38 | (11.5) | 1 | |
| Yes | 171 | (34.2) | 33 | (19.3) | **1.66 [1.06–2.60]** | **0.03** |
| **Pregnancy monitoring during first lockdown** | | | | | |
| Having an unmet need to communicate with health professionals about course of pregnancy/ childbirth during pandemic | | | | | | |
| No | 295 | (59.0) | 29 | (9.8) | 1 | |
| Yes | 205 | (41.0) | 42 | (20.5) | **1.66 [1.08–2.55]** | **0.02** |
| Took medication for mood disorders/sleeping disorders | | | | | | |
| No, because I did not need it | 456 | (91.2) | 50 | (10.9) | 1 | |
| Yes | 20 | (3.9) | 8 | (40.0) | 1.77 [0.87–3.61] | 0.11 |
| No, but I would have liked to | 24 | (4,9) | 13 | (54.2) | **2.86 [1.74–4.71]** | **<0.0001** |

* Weighted and rounded values using Newton's algorithm for discrete or qualitative variables [21]. For continuous variables (age, pregnancy term), mean (standard deviation) were presented.

** Adjusted Prevalence Ratio (aPR), Confidence Interval 95% (95%CI) and p-value obtained with robust variance Poisson regression model.

[a] Anxiety symptoms assessed two months after the first lockdown ended using to Hospital Anxiety and Depression scale (HAD). Score of >10 for all seven anxiety questions were considered to have anxiety symptoms (see definition of the variable of interest in Methods section).

[b] At time of study (July 2020).

[c] Women on maternity leave and unemployed women were classified according to their current SPC category or their most recent category prior to ending work respectively.

[d] Gestational diabetes, Pre-eclampsia, Preterm labour, Gestational hypertension, etc.

[e] Body Mass Index$\geq$25kg/m$^2$.

2 virus was a protective factor. A majority of participants reported heightened feelings of powerlessness, frustration or fear during lockdown. Regarding anxiety symptom frequency, measured two months after the first lockdown ended (July 2020), 14.2% of the respondents had anxiety symptoms. The following factors were associated with increased anxiety symptom frequency: a pregnancy-related pathology, overweightness/obesity, a friend or family member who had been diagnosed with Covid-19 or had symptoms suggestive of the disease, perceiving little or no social support during the lockdown, having one or more children under the age of six in the household, unsuccessful attempts to have an exchange with a healthcare professional about their pregnancy or about the hospitalisation process for childbirth, and no access to medication to treat mood disorders or sleep disorders.

The present study highlights that the first lockdown in France was associated with a deterioration of self-perceived mental health in one of five pregnant women. Moreover, compared to women of childbearing age (18–49 years) participating in a survey–CoviPrev- conducted in the French general population at the same time [31], this deterioration appeared significantly more frequent among pregnant women (21.1% vs 7.7%; p<0.0001) (unpublished data). This result is consistent with a recent literature review by Brooks *et al.* on the negative psychological impact of previous quarantines [32]. Those authors also pointed out that providing sufficient and accurate information to people self-isolating because of a pandemic gave them a better understanding of the reasons and context, and consequently played a key role in preventing the harmful psychological effects of lockdown. Likewise, a Chinese study on 1873 pregnant women showed that having access to information provided by hospital services on antenatal care during the Covid-19 pandemic was a protective factor against stress, anxiety and depression [33]. In the present study, we also showed that women with a good level of knowledge about the modes of transmission of the virus were less likely to have psychological deterioration.

The frequency of anxiety symptoms measured two months after the lockdown in our study sample was 14.2%. Knowing the variations in prevalence between countries and the screening tools used, this frequency was of the same order of magnitude as that observed internationally in pregnant women (22.4%) [34] before the current pandemic and among women of childbearing age in France and more generally in Europe (20.2% according to the French *Baromêtre Santé* survey 2017 (unpublished data); 14.9% according to the European Study of the Epidemiology of Mental Disorders-ESEMeD—2001–2003 [35]). An eleven-week study conducted in the United Kingdom showed that the GAD-7 anxiety score in 11 pregnant women infected with SARS-CoV-2 rose to a peak at the beginning of the first lockdown when the number of deaths in the UK during the study period was at its highest, and subsequently progressively *decreased* over the rest of the study period [36]. While one might suppose that a similar peak occurred in France, our results for anxiety—obtained two months after the first French lockdown ended—may reflect adaptation and habituation to Covid-19 health risks by pregnant women, something already observed in women of childbearing age in the general population [37,38]. This hypothesis is strengthened by the fact Covimater was conducted in July 2020, when the monitoring markers for the pandemic could be perceived as reassuring by the women in France.

One interesting finding in our study, is that the proportion of those defined with anxiety symptoms was significantly lower than in women of childbearing age participating in CoviPrev at the same time (14.2% *vs* 24.8%; p<0.0001). This echoes a finding in a case-control survey conducted in the Turkish context [39]. However, although we weighted out data, a more in-depth analysis—taking into account socio-demographic variables and general health status—is needed to better explain these findings.

Anxiety symptom frequency in Covimater was significantly higher in participants with a pregnancy-related pathology, those who were overweight or obese, and those with a loved one

who had been diagnosed with Covid-19 or had symptoms suggestive of the disease. Information on Covid-19 morbidity and mortality and on populations at risk of severe forms of the disease—in particular obese patients—were widely diffused through the media and social networks at the beginning of the pandemic in France. This may have generated stress and increased anxiety in the populations concerned. Furthermore, pregnancy-related pathologies, overweight and obesity have all been associated with poorer perinatal mental health, and in particular, anxiety [40,41]. In another Turkish study of pregnant women during the SARS-CoV-2 pandemic, obesity was one of the principal risk factors of higher anxiety and depression scores [42]. More generally, poor physical health or chronic illness are associated with an increased risk of perinatal psychiatric disorders [43].

Our analyses also show that having one or more children under six years of age during the first French lockdown was associated with higher anxiety symptom frequency. Several studies have pointed out that having young, less autonomous, non-school going children is an additional source of daily stress during the current pandemic, especially for mothers must also continue their professional activity during lockdown. For instance, in an internet survey of 10165 women in the UK, those who lived with young children under five years old had significantly higher General Health Questionnaire (GHQ-12) mental distress scores [44]. Similarly, a Canadian longitudinal study conducted before and during the SARS-CoV-2 pandemic showed an increase in anxiety score in women who experienced difficulties with childcare [45].

Some factors were associated with both psychological deterioration during the lockdown and anxiety two months after, for instance perceiving little or no social support. This finding reflects the international literature on the importance of quality social and familial support in maintaining good physical and mental health, in particular for people experiencing especially difficult periods of vulnerability such as pregnant women [46,47]. Other studies conducted during the ongoing pandemic have highlighted the importance of perceived social support in pregnant women's well-being, and its inverse association with anxiety [15,48,49]. One study highlighted that loneliness mediates both the perceived social support-anxiety relationship and the social support-depression relationship [48].

Other factors associated with higher frequency of anxiety symptoms during the first lockdown were related to unsuccessful attempts to have an exchange with healthcare professionals about their pregnancy and an unmet need for medication for mood or sleep. The lockdown has resulted in less access to care, both in the general population and for women in the perinatal period ([26,50]). Thus, use of emergency psychiatric care drastically decreased in France [51]. In the United States (Massachusetts), 35.9% of pregnant women reported a lower access to mental healthcare [52]. Women who had not already started psychiatric care may have had difficulty seeking information or treatment from professionals. Before the pandemic, the unmet need for antenatal care, particularly mental health care, was already a public health problem in France. Indeed, several studies conducted before the current health crisis estimated that only a quarter of pregnant women with mental health disorders receive specialised care [53,54].

To the best of our knowledge, Covimater was the first national study in France that explores the experiences, behaviours and mental health of pregnant women during the SARS-CoV-2 pandemic. In order to compare results, the methodology and some of the questions were the same as those used in another French survey—CoviPrev—which was conducted in the general population at the same time. Compared to other international studies that often focused on third trimester pregnant women, Covimater's design included women with different gestational ages during the first lockdown. In Covimater, although some groups compared were unbalanced in size (with consequently reduced power), this did not prevent the identification of significant associations with the variable of interest. Covimater had some limitations. First,

even though the study has a consistent internal validity, the use of a panel and quota sampling imply that these findings lack external validity (cannot be generalized to the whole French population of pregnant women). However, no alternative method would have allowed this study to take place such a short time after the lockdown, thus avoiding a significant recall bias. Second, sampling bias could explain the overestimation of the percentage of pregnant women with chronic diseases or obesity. Third, as the study questionnaire was self-administered including some retrospective assessments, there is a risk of potential social desirability and recall biases. However, there is no reason that the latter should be limited to the sub-group of pregnant women.

## Conclusion

The results of this study emphasize the need of more adapted support for pregnant women during the SARS-CoV-2 pandemic, in particular: i) exchanges with health professionals about their pregnancy and hospitalization for childbirth, ii) psychological support, iii) increased information about the modes of transmission of the virus, iv) social support to avoid social isolation and v) childcare services or a limitation of school closures. Strategies to prevent perinatal psychiatric disorders are essential (i) to limit the negative impacts of impaired mental health on the course of their pregnancy, (ii) to ensure the mother-newborn bond is established, and (iii) to guarantee appropriate development of the infant/child.

## Acknowledgments

Our thanks to Dorothée Lamarche (BVA group) for her invaluable help in creating the study questionnaire and to Jude Sweeney (Milan, Italy) for the English revision and copyediting of this manuscript.

## Author Contributions

**Conceptualization:** Alexandra Doncarli, Lucia Araujo-Chaveron, Imane Khireddine, Nolwenn Regnault.

**Formal analysis:** Alexandra Doncarli.

**Methodology:** Alexandra Doncarli, Lucia Araujo-Chaveron, Nolwenn Regnault.

**Validation:** Nolwenn Regnault.

**Writing – original draft:** Alexandra Doncarli, Lucia Araujo-Chaveron.

**Writing – review & editing:** Catherine Crenn-Hebert, Marie-Noëlle Vacheron, Christophe Léon, Imane Khireddine, Francis Chin, Alexandra Benachi, Sarah Tebeka, Nolwenn Regnault.

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
