## [Decision Letter · Decision Letter 0]

24 Feb 2023

Mental health of pregnant women during the SARS-CoV-2 pandemic in France: evolution of self-perceived psychological state during the first lockdown, and anxiety frequency two months after the lockdown ended.

PONE-D-22-19614

Dear Dr. Duncali,

We’re pleased to inform you that your manuscript has been judged scientifically suitable for publication and will be formally accepted for publication once it meets all outstanding technical requirements.I congratulate you for your work and thank you for the patience to wait until the full peer-review examinations were due.Please note that I have acted as reviewer 3 in your paper.

Kind regards,

Silva Ibrahimi, PhD

Academic Editor

PLOS ONE

Journal Requirements:

1. Please provide additional details regarding participant consent. In the ethics statement in the Methods and online submission information, please ensure that you have specified (1) whether consent was informed and (2) what type you obtained (for instance, written or verbal, and if verbal, how it was documented and witnessed). If your study included minors, state whether you obtained consent from parents or guardians. If the need for consent was waived by the ethics committee, please include this information.

Reviewers' comments:

Reviewer's Responses to Questions

**Comments to the Author**

1. Is the manuscript technically sound, and do the data support the conclusions?

Reviewer #1: Yes

Reviewer #2: Yes

Reviewer #3: Yes

2. Has the statistical analysis been performed appropriately and rigorously? 

Reviewer #1: Yes

Reviewer #2: Yes

Reviewer #3: Yes

3. Have the authors made all data underlying the findings in their manuscript fully available?

Reviewer #1: Yes

Reviewer #2: No

Reviewer #3: Yes

4. Is the manuscript presented in an intelligible fashion and written in standard English?

Reviewer #1: Yes

Reviewer #2: Yes

Reviewer #3: Yes

5. Review Comments to the Author

Reviewer #1: Data results support paper conclusions, paper is technically sound and very interesting focusing on very important problems post-pandemic, statistic was performed appropriately. Very important is that limitations have been described and discussed.

Reviewer #2: I congratulate the authors on a scientifically sound paper that will make a tremendous impact in the way we approach mental health both from a clinical and public health perspective, not just in France but globally as well.

Reviewer #3: I congratulate the authors for their interesting and very sound research! The statistical analysis and data interpretation give the research the impact of a well-designed and significant work. The research methodology and design have also addressed all issues.

6. PLOS authors have the option to publish the peer review history of their article (what does this mean?). If published, this will include your full peer review and any attached files.

Reviewer #1: No

Reviewer #2: No

Reviewer #3: No

---

## [Editor Report · Acceptance letter]

13 Apr 2023

PONE-D-22-19614 

Mental health of pregnant women during the SARS-CoV-2 pandemic in France: evolution of self-perceived psychological state during the first lockdown, and anxiety frequency two months after the lockdown ended. 

Dear Dr. Doncarli:

I'm pleased to inform you that your manuscript has been deemed suitable for publication in PLOS ONE. Congratulations! Your manuscript is now with our production department. 

Kind regards, 

on behalf of

Dr. Silva Ibrahimi 

Academic Editor

PLOS ONE